# Synthesis, Characterization, and Antibacterial Activity of Ag_2_O-Loaded Polyethylene Terephthalate Fabric via Ultrasonic Method

**DOI:** 10.3390/nano9030450

**Published:** 2019-03-18

**Authors:** Armin Rajabi, Mariyam Jameelah Ghazali, Ebrahim Mahmoudi, Amir Hossein Baghdadi, Abdul Wahab Mohammad, Nadia Mohd Mustafah, Htwe Ohnmar, Amaramalar Selvi Naicker

**Affiliations:** 1Centre for Materials Engineering and Smart Manufacturing, Faculty of Engineering and Built Environment, Universiti Kebangsaan Malaysia, 43600 UKM Bangi, Selangor, Malaysia; baghdadi.amirhossein@gmail.com; 2Department of Chemical and Process Engineering, Faculty of Engineering and Built Environment, Universiti Kebangsaan Malaysia, 43600 UKM Bangi, Selangor, Malaysia; ebi.dream@gmail.com (E.M.); wahabm@eng.ukm.my (A.W.M.); 3Department of Rehabilitation Medicine, Faculty of Medicine, Universiti Teknologi MARA, 47000 Sg. Buloh, Selangor, Malaysia; nadiamustafah@gmail.com; 4Rehabilitation Medicine Unit, Department of Orthopaedics & Traumatology, Faculty of Medicine, University Kebangsaan Malaysia, 56000 Cheras, Kuala Lumpur, Malaysia; htwe.om@gmail.com (H.O.); asnaicker@yahoo.com (A.S.N.)

**Keywords:** ultrasonic synthesis, Ag_2_O, morphology, polyethylene terephthalate (PET) fabric, inhibition zone

## Abstract

In this study, Ag_2_O was synthesized on polyethylene terephthalate fabrics by using an ultrasonic technique with Ag ion reduction in an aqueous solution. The effects of pH on the microstructure and antibacterial properties of the fabrics were evaluated. X-ray diffraction confirmed the presence of Ag_2_O on the fabrics. The fabrics were characterized by Fourier transform infrared spectroscopy, ultraviolet–visible spectroscopy, and wettability testing. Field-emission scanning electron microscopy verified that the change of pH altered the microstructure of the materials. Moreover, the antibacterial activity of the fabrics against *Escherichia coli* was related to the morphology of Ag_2_O particles. Thus, the surface structure of Ag_2_O particles may be a key factor of the antibacterial activity.

## 1. Introduction

At present, microbial contamination in healthcare facilities, especially in hospitals, has caused considerable concern among people. Infectious diseases have become a critically important global healthcare problem, which may cause excessive loss of money and human lives [1,2]. Textile materials have become the main base of cross infections in hospitals and medical institutions with the increasing occurrences of antibiotic-resistant bacterial strains and community-type outbreaks. Polyethylene terephthalate (PET) can be utilized as a reusable protective textile in hospitals due to its excellent mechanical properties, such as thermal stability, durability and reusability, ease of processing, and cost-effectiveness [3,4]. The production and consumption of reusable PET in medical applications, such as lab coats and privacy drapes, have constantly increased in recent years due to the increase in demand for reusable textiles. However, PET is vulnerable to contamination of many microorganisms, such as viruses, bacteria, and spores; some of these may survive for approximately 90 days on PET [5,6]. Thus, contaminated PET can serve as an important medium for infectious diseases. For this reason, effective and generable antimicrobial PET fabrics should be investigated and developed. At present, nanoparticles (NPs) have been recognized as improving antibacterial activity and have attracted considerable attention from scholars in their applications, such as clothing, wound dressings, food packaging, cosmetics, and dental restorative materials [7,8,9]. On this basis, PET textures treated with antimicrobial agents have been generally investigated and the results have shown its capacity to halt the development of pathogenic microorganisms, such as parasites and microscopic organisms [10,11]. Antimicrobial NPs present many unique benefits compared with popular antibacterial agents. These benefits include cost-effectiveness and overcoming resistance [12]. Several antimicrobial agents (zinc, copper, and silver) have been utilized as antimicrobial varnishes on fabric materials [13,14,15]. Among the previously mentioned antibacterial materials, silver nano-oxide has been extensively used on cotton and artificial fibers, such as thin polymer films, wound pads, polyester, and drinking water-related applications [16,17]. Notably, the antimicrobial nature of nano-Ag_2_O has been widely recognized due to its wide spectrum killing and moderately low risk to microbial resistance development [18,19]. Silver’s antibacterial properties are due to its high valence state which causes strong electrostatic attractions between the Ag ion and bacteria [14,20,21]. The antimicrobial activity of metallic NPs is recognized to be remarkably influenced by their shape and size, which are largely affected by preparation procedures. Several matters occur in the synthesis of NP-coated fabrics. The remaining matters are those with expensive and advanced equipment requirements and the difficulties of control methodology.

Ultrasonication is an effective method used to synthesize inorganic materials, counting metals, and metal oxides [22,23]. Sonochemistry is derived from acoustic cavitation, which is the creation, growth, and implosive collapse of bubbles within a liquid. Acoustic cavitation is the main mechanism following the sonochemical synthesis of nanomaterials. The development and collapse of bubbles inside the solution are related to the applied ultrasound frequency [24,25]. Other benefits of sonochemical synthesis include convenience, environmental-friendliness, rapidity instead of fast processes, and effectiveness [26,27]. Jimmy et al. [28] have reported that photocatalytically active TiO_2_ NPs synthesized through an ultrasonic method are more profitable than that of commercial NPs, owing to the improved oxide crystallinity with fast hydrolysis rate by ultrasound. On this basis, a high-intensity (30 kHz) ultrasound sonochemical method has been used to synthesize and recognize novel nanomaterials without using high temperatures and long reaction times [29].

However, only a few studies have focused on the synthesis of Ag_2_O nanomaterials by using an ultrasonic technique. This study aimed to synthesize Ag_2_O nano-loaded PET fabrics under various pH by using an ultrasonic method and to characterize the microstructure and antibacterial behavior of the fabrics.

## 2. Materials and Methods

AgNO_3_ (99% *w*/*w*) and NaOH pellets (98% *w*/*w*) were obtained from a local brand company (R&M, Semenyih, Malaysia). One mole of AgNO_3_ was dissolved in deionized water under magnetic stirring for 15 min at room temperature and pH solution was adjusted by sodium hydroxide (Table 1). PETs 5 mm in diameter were obtained from a local brand, washed with distilled water, and dried in an air oven at 70 °C for 12 h. PETs were soaked in the solutions for 5 h and were mildly stirred at 25 °C. The mixture consisted of aqueous solution and PETs were sonicated (30 kHz) for 15 min. PETs were removed from the solution, washed with distilled water, and dried in an oven at 70 °C for 12 h. X-ray diffraction (XRD) analysis was conducted on PETs using a Bruker AXS (GmbH Oestliche Rheinbrueckenstr, Karlsruhe, Germany) diffractometer with monochromatic Cu–Kα radiation (λ = 0.1541 nm) at 4 mA and 40 kV. A Perkin–Elmer Spectrum 400 Fourier transform infrared (FTIR/FTNIR) spectroscope (Akron, OH, USA) was used to evaluate the chemical composition of the samples in the region 500–4000 cm^−1^. The UV–Vis absorption properties of the samples were evaluated through absorption spectroscopy using a Perkin–Elmer Lambda-35 UV–Vis spectrophotometer. Prior to electron microscopy, the samples were coated with a thin layer of iridium to avoid the charging effect instigated by the non-conductive property of the fabrics. The morphological behavior and contact angle (CA) of the samples were assessed using a field emission scanning electron microscope (FESEM; Zeiss Merlin, Zurich, Switzerland) and an optical tensiometer (Theta of Attension, Biolin Scientific, Hängpilsgatan, Västra Frölunda, Sweden). Disk diffusion antibiotic sensitivity testing (Kirby–Bauer antibiotic testing) was employed to evaluate the efficiency of PETs against *Escherichia coli*. Firstly, fabrics embedded with nanoparticles at different pH were a cut as disks with a diameter of 0.5 mm. Then, 30 μL of *E. coli* solution (nutrient broth Himedia M001, West Chester, PA, USA) with a concentration of 16 × 10^7^ cells/mL (OD600 of 0.2) was spread on a petri dish containing nutrient agar (Himedia M002, West Chester, PA, USA). The fabric discs containing Ag2O nanomaterial were placed on each Petri dish and a piece of uncoated fabric was used as a control in each set of tests. The Petri dishes were incubated at 35 °C for 24 h, digital images of the plates were captured, and the inhibition zone was calculated using an image processing software (image J.140, University of Wisconsin, Madison, WI, USA) [20,30]; each test was repeated three times and the average results were reported.

## 3. Results and Discussion

### 3.1. X-ray Diffraction Analysis

The XRD results of presence and purity of Ag_2_O crystalline grown on the surface of PETs are shown in Figure 1. As shown in Figure 1, the characteristic peaks of a crystalline PET structure in the fabric appeared at peaks 2θ = 16°, 22°, and 26° for the original and coated fabrics. However, Ag_2_O-PET fabric samples showed characteristic diffraction peaks of Ag_2_O at 2θ = 26.9°, 32.69°, 37.94°, 54.9°, 65.54° and 69°, which agrees with silver (I) oxide FCC crystalline phase (JCPDS 041-1104). Meanwhile, these Ag_2_O peaks were clearly absent in the F1 sample due to the low quantity of materials, as previously reported for other materials [20,31]. Furthermore, no strange diffraction peaks were detected in the XRD patterns of all Ag_2_O-coated fabric samples, which confirms the existence of the high crystalline quality of the Ag_2_O-coated fabric surface. For the main diffraction peak (2θ = 32.69°), the full width at half maximum (FWHM) became sharp and narrow for the F3, F4, and F5 samples, where the values are 0.4379075°, 0.4278235°, and 0.4119844°, respectively, which shows the increase in the crystallinity of Ag_2_O nanoparticles for the F4 and F5 samples with an increase in pH. The average crystalline sizes of Ag_2_O NPs at different pH (6.5, 7.5, 8.5, 9.5, and 10.5) were calculated using Scherrer’s equation [31,32] and were found to be 19.23 nm, 20.65 nm, 18.91 nm, 19.35 nm, and 21 nm, respectively. The crystalline size of the materials moderately increased from F4 into F5 and reached the maximum value with the highest pH. However, the decrease in crystalline size of Ag_2_O NPs for F3 was confirmed by the inhibition of peak intensity and increase of FWHM. The increment in the crystalline size was due to high nucleation rate and growth of Ag_2_O nanocrystallites with the accessibility of sufficient thermal energy due to high NaOH concentration in the precursor, as has been reported in a previous works [20,33].

### 3.2. Microstructural Observation

The size and morphology of PET fabrics after Ag_2_O coating can be clearly observed at two different magnification sizes, as shown in Figure 2. The microstructures of the synthesized materials at low magnification are similar and agglomeration particles can be observed. Although the growth mechanism of Ag_2_O is difficult to determine, the details for the formation of different particle sizes may be explained based on the literature [34,35]. In acid solution (pH = 6.5), a small particle of nano-sized Ag_2_O on the PET surface was observed, as shown in Figure 2a by the yellow circle. This condition was ascribed to the etching in acidic solution. The concentration of H^+^ ions increases the reaction with OH^−^ on the surface of the Ag^+^ precursor and remarkably inhibits the growth of Ag_2_O. However, the yellow circle (Figure 2b) reveals a starfish-like morphology at high magnification, which is composed of a series of regular rod shapes in the range of 200–4500 nm and 160–200 nm for length and width, respectively. Notably, the effect of H^+^ ions is inferior, and Ag_2_O particles start growing from the Ag^+^ precursor at high pH. Therefore, the role of pH can be crucial in controlling the size and shape of NPs. Figure 2d (pH = 7.5) at high magnification shows that the powder morphology involves a large amount of semiregular spherical particles with sizes in the range of 100–400 nm. Cavitation bubbles were generated and oscillated with high frequency with the introduction of ultrasound [36,37]. The oscillation of cavitation bubbles fragmented the microstructure of the rods into small grains and were moved by acoustic streaming. Most recently, sonofragmentation of particles by a shock wave has been added to the effects of a shock wave created by ultrasound irradiation. Zeiger et al. [38] have successfully eliminated many possible mechanisms of particle breakage, such as particle–particle, particle–horn, and particle–wall collisions through kinetics. Consequently, they inferred breakage by a particle shock wave interaction as a viable mechanism for the sonofragmentation of acetylsalicylic acid. In this study, particles dispersed in the liquid phase, stuck to one another, and formed an agglomerate structure, which resulted in poor access to the constituent rod-like particles in the powders (Figure 2d). The microstructure of the powders at pH = 8.5 consisted of two different morphologies, namely, semiregular spherical and rod-shaped particles (Figure 2f). The growth mechanism of the Ag_2_O rod-shaped particles was not known at this stage. However, sonication irradiation at pH = 8.5 facilitated the incorporation of rod-shaped-inducing fragmented rods (nucleation sites) into semiregular spherical and inducing rod-shaped particles (F3). In fact, ultrasound-induced interparticle collisions led to a considerable enhancement of intergrain coupling. Notably, the moderate change from rod-shaped particles to semiregular spherical particles was observed in the morphology in Figure 2h. However, the rod-shaped particles were shortened to semi-spherical due to their lower Gibbs free energy and became homogenously dispersed, as shown in Figure 2j.

### 3.3. FTIR Results

FTIR has become a popular method used to obtain data on functional groups. The FTIR spectra in the 500–4000 cm^−1^ region of the original PET and Ag_2_O-coated fabrics are displayed in Figure 3. FTIR shows two regions, namely, the functional group (4000–1504 cm^−1^) and fingerprint regions (1504–500 cm^−1^). As shown in Figure 3, all the functional groups of the original PET and Ag_2_O-coated fabrics are similar. Therefore, the main structure of the materials is the same. The summary of infrared vibration is represented in Table 2 [39,40,41]. Although the transmittance spectra appear relatively similar to each other, a decrease in the intensity of the F3 sample can be observed, which is attributed to particle size/morphology, as discussed in the FESEM image (Figure 2).

### 3.4. UV–Vis Tests

Figure 4 displays the UV–Vis absorption spectrum of the obtained samples at different pH. The maximum peak of PET shifted to left with Ag_2_O loading, which verifies the existence of Ag_2_O on the fabric (F2, F3, F4, and F5), as shown in the XRD results. The low peak intensity in the F2 sample (236 nm) may be due to low quantity of Ag_2_O, as discussed in the XRD results (Figure 1). Furthermore, a sharp absorption band was noted at 237 nm, 235 nm, 234 nm, 232 nm, and 230 nm for the Ag_2_O-loaded fabric at different pH values of 8.5, 9.5, 10.5, 7.5, and 6.5, respectively. The increased absorption of the samples at pH = 8.5 compared with the rest is due to the rod morphology of NPs, as shown in the FESEM images (Figure 2). Rokade et al. [42] have reported that the change of absorption bands is related to morphology of particles, such as Ag_2_O nanopowders.

### 3.5. Wettability Results

Hydrophobicity and hydrophilicity of materials, that is, surfaces with contact angles (CA) higher and lower than 65°, respectively, have been obtained based on captured photos and contact angle measurements of 1 mL water droplets as a quantitative method [20,43]. Based on the literature [44,45], surface energy and surface roughness are the main factors used to control the CA. Therefore, wettability of the surfaces is changed based on the modifications of roughness and morphology of microstructures. A typical PET fabric can be completely wetted by water droplets (Figure 5a) due to the presence of hydrophilic –OH groups, as illustrated in Table 2. The loading of Ag_2_O particles onto the fabric could make the fabric rough and hydrophobicity would be introduced to the PET fabric. Here, the average CA values of PET-loaded Ag_2_O at different pH values (6.5, 7.5, 8.5, 9.5 and 10.5) were 95.2°, 94.1°, 122.1°, 94.8° and 96.9°, respectively, and uncoated had a value of ~0° (Figure 5). As shown in Figure 5, the F3 sample exhibits high CA and is more hydrophobic than other samples, which confirms that the surface was successfully modified and has obtained a self-cleaning condition. Therefore, high hydrophobicity (CA of approximately 122.1° similar to Teflon coating [46]) of the PET fabric can exhibit good self-cleaning, and is highlighted as a promising candidate for antibacterial activity. Suryaprabha et al. [47] have reported that the self-cleaning and antibacterial properties of fabrics are improved with an increase in CA.

### 3.6. Antibacterial Tests

The inhibition zone of the obtained PET-loaded Ag_2_O against *E. coli* is shown in Figure 6, and the values are given in Table 3. The diameter of the inhibition zone of the F3 sample is higher than that of other samples. This result suggests that the F3 sample with nanorod morphology exhibits a remarkable growth inhibitory effect against *E. coli*. The results of the present study are similar to results reported by other researchers [20,48,49]. The observed good antibacterial activity is attributed to (a) the large surface area of Ag_2_O nanorods as seen in FESEM analysis and (b) the high absorption behavior of Ag_2_O nanorods as seen in UV tests. The possible mechanisms for the antibacterial activity of Ag_2_O are expressed as follows: (i) Ag_2_O contacts and penetrates bacterial cells due to its nanometre scale and rod-like morphology. DNA misses its repetition capability when bacteria are subjected to Ag_2_O NPs. Cells are influenced by oxidative stress, which is caused by the inhibition of ATP synthesis [50,51] (the energy currency of cells for all organisms) and occurrence of reactive oxygen species (ROS) [20,52]. ROS increases with the increase in surface area and appropriate crystal sizes. XRD results illustrate that the crystalline size of F3 is 18.91 nm, which can support antibacterial activity. (ii) After the penetration of nanomaterials, the bacterial enzymes deactivate by releasing atomic Ag^0^ and ionic Ag^+^ clusters, which kill the bacterial cells and increase the inhibition zone [53,54].

## 4. Conclusions

In this study, Ag_2_O-loaded PET fabrics were successfully synthesized using an ultrasonic method with AgNO_3_ as a precursor solution. The morphology, CA, and antibacterial activity of the fabrics were dependent on the change of pH. A sharp band at 237 nm in the UV test with the lowest transmittance in the FTIR results was observed at pH 8.5 with nanorod morphology, as shown by FESEM analysis. Furthermore, the antibacterial results revealed a direct relationship with morphology. Thus, the maximum inhibition zone obtained for the fabrics was 14 ± 2 mm because of the large surface area of the Ag_2_O nanorods. The results of this study may be applicable to medical devices that are coated with Ag_2_O nanorods via ultrasonic methods to improve antibacterial activity.

## Figures and Tables

**Figure 1 nanomaterials-09-00450-f001:**
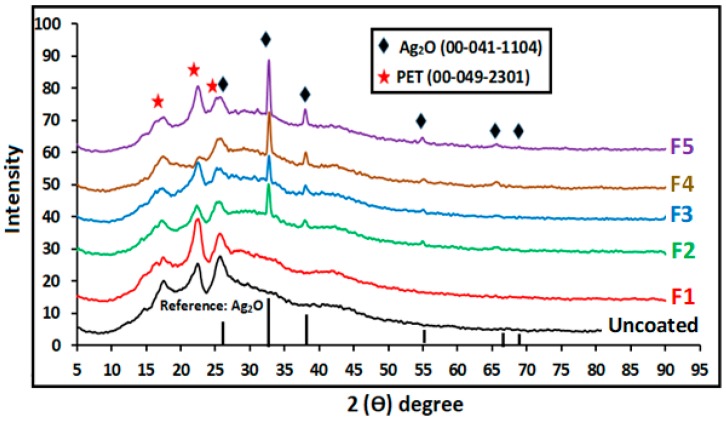
X-ray diffraction (XRD) patterns of synthesized Ag_2_O with different pH.

**Figure 2 nanomaterials-09-00450-f002:**
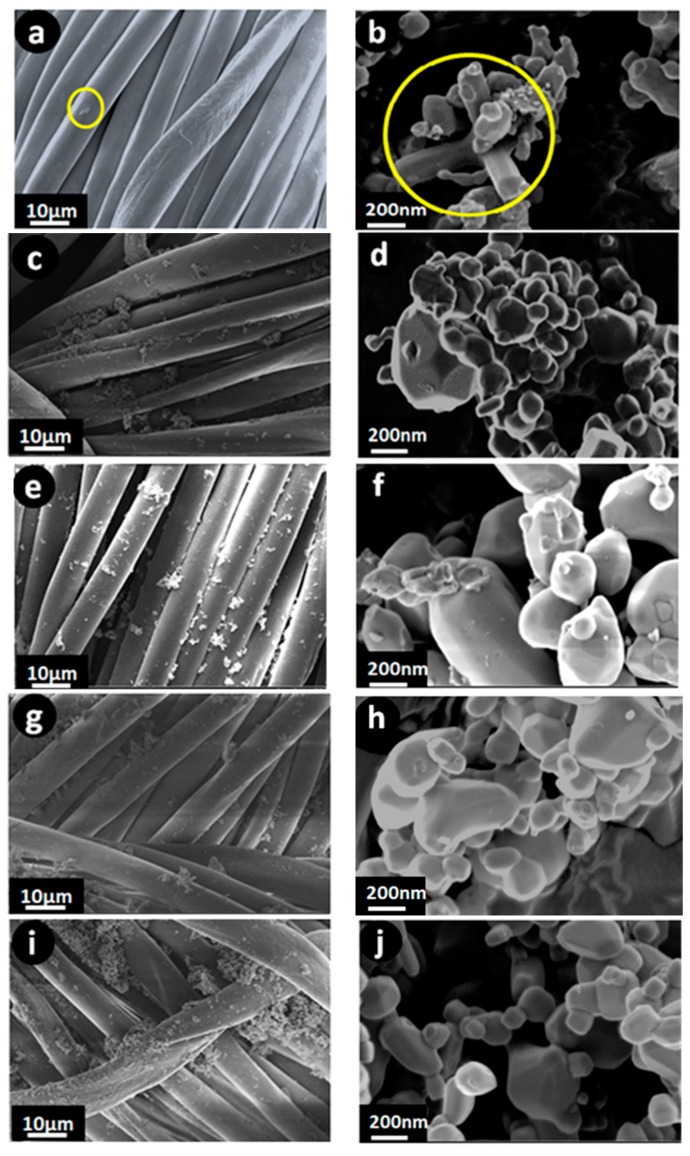
Field emission scanning electron microscope (FESEM) images of the samples at different magnifications. (**a**,**b**) F1, (**c**,**d**) F2, (**e**,**f**) F3, (**g**,**h**) F4, (**i**,**j**) F5, and (**k**,**l**) uncoated.

**Figure 3 nanomaterials-09-00450-f003:**
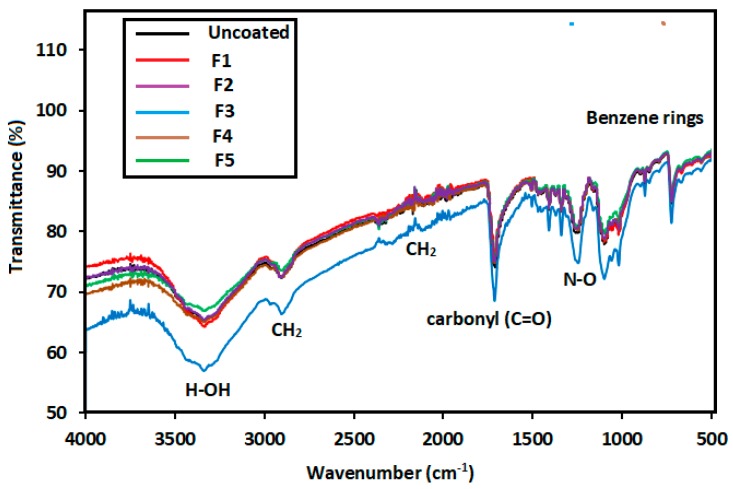
Fourier transform infrared (FTIR) spectra of the samples at different pH.

**Figure 4 nanomaterials-09-00450-f004:**
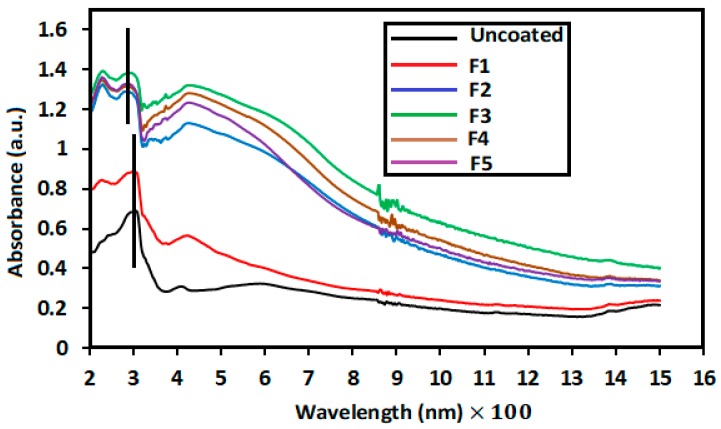
UV–Vis absorbance spectra of the samples.

**Figure 5 nanomaterials-09-00450-f005:**
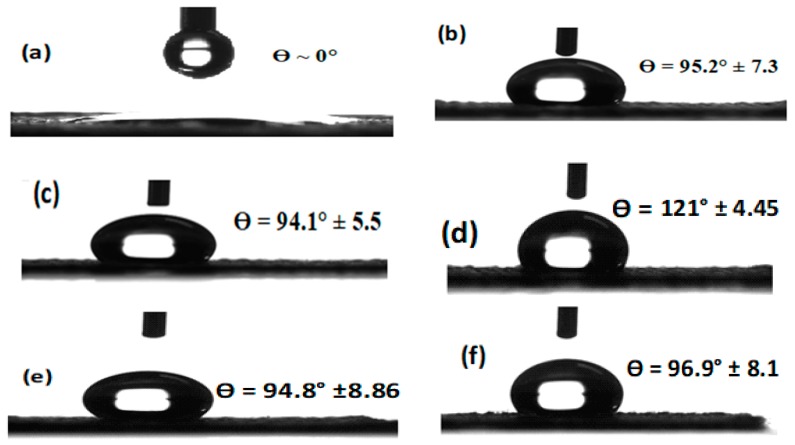
Contact angle of the samples. (**a**) Uncoated, (**b**) F1, (**c**) F2, (**d**) F3, (**e**) F4, and (**f**) (F5).

**Figure 6 nanomaterials-09-00450-f006:**
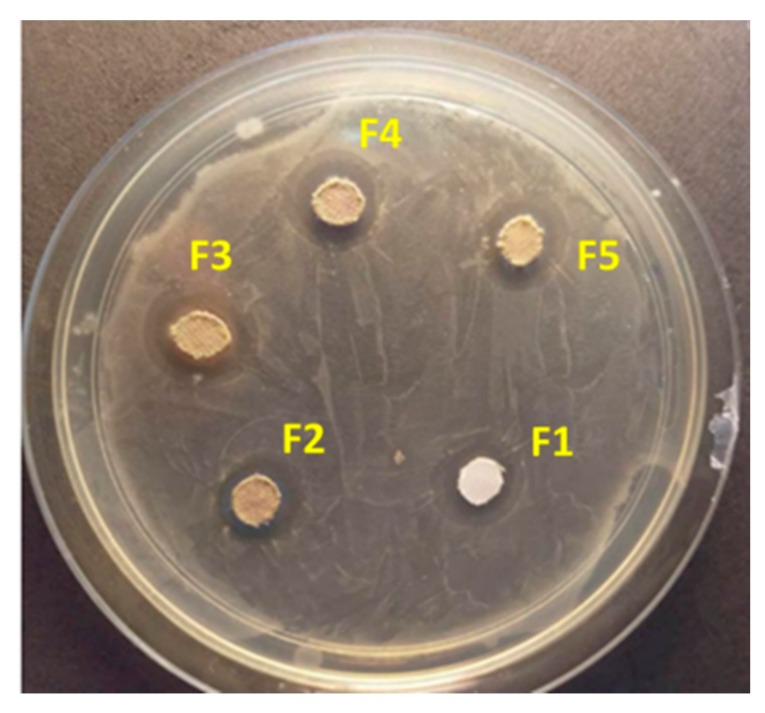
Antibacterial activity of Ag_2_O-loaded PETs.

**Table 1 nanomaterials-09-00450-t001:** Sample label with different pH. Legend: PET, polyethylene terephthalate.

Sample Label	Description
F1	Ag_2_O-coated PET fabric synthesized at pH 6.5
F2	Ag_2_O-coated PET fabric synthesized at pH 7.5
F3	Ag_2_O-coated PET fabric synthesized at pH 8.5
F4	Ag_2_O-coated PET fabric synthesized at pH 9.5
F5	Ag_2_O-coated PET fabric synthesized at pH 10.5

**Table 2 nanomaterials-09-00450-t002:** Infrared vibration from FTIR analysis.

Wavenumber (cm^−1^)	Vibration Characteristics	Wavenumber (cm^−1^)	Vibration Characteristics
3339	–H bonded OH stretch	1339	N–O stretching
2874	C–H_2_ groups	1247	C–O) vibrations
2361	C–H_2_ groups	1098	(C–O) vibrations
1712	carbonyl (C=O) stretching	1018	(C–O) vibrations
1560	phenyl groups	650–900	benzene rings
1408	methylene groups		

**Table 3 nanomaterials-09-00450-t003:** Antibacterial activity of the samples.

Sample Label	Inhibition Zone Diameter (mm)
F1	11 ± 0.3
F2	13 ± 0.1
F3	14 ± 0.2
F4	12 ± 0.3
F5	11 ± 0.3

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
