# Peer review of "Synthesis, Characterization, and Antibacterial Activity of Ag2O-Loaded Polyethylene Terephthalate Fabric via Ultrasonic Method"

_nanomaterials, 2019, doi:10.3390/nano9030450_

Reviewer 1 Report

Dear Authors,

the paper deals with a very interesting topic. It is well written and well organized. The English is very fluent.

I suggest to add some literature references for the following sentence:

  "However, few studies have focused on the synthesis of Ag2O nanomaterials by using an ultrasonic technique".

I would like to know if you used any internal standard in XRD measurements.

Best regards

Author Response

Question:

I suggest to add some literature references for the following sentence:

 "However, few studies have focused on the synthesis of Ag2O nanomaterials by using an ultrasonic technique".

Arthurs reply

Thanks for the suggestions. The following references has been added to the article

Question:

I suggest to add some literature references for the following sentence:

 "However, few studies have focused on the synthesis of Ag2O nanomaterials by using an ultrasonic technique".

Arthurs reply

Thanks for the suggestions. The following references has been added to the article

Reviewer 2 Report

Rajabi, Ghazali and coworkers present a manuscript in which synthesis and characterization of  Ag nanoparticles-loaded PET fabric is presented. Ag2O was deposited onto the cloth via ultrasonication and the effect of different pH adopted during synthesis is investigated by mainly chemical methods with a small antibacterial activity test. The possible application field of the generated clot is antibacterial textiles in healthcare systems. The novel approach for Ag2O synthesis via ultrasounds is innovative, simple and fast, providing the field with a new tool.

Point 1:

The introduction highlights the need for reusable cloth that display antibacterial activity especially for clinical settings, where multidrug resistant strains are spreading. Then metals use as antimicrobial agents is introduced. As for other parts, the authors should carefully revise and thoroughly improve biological aspects. Concerning the introduction, an improvement in the papers cited is suggested. More relevant research in the field should be acknowledged (e. g. Levy & Marshall, 2004; O’Neill, J. Tackling Drug-Resistant Infections Globally: Final Report and Recommendations; Pelgrift & Friedman, 2013; Mijnendonckx et al., 2013; Lemire et al., 2013).

Point 2:

The Materials and Methods section should be rephrased in a clearer way in its whole, especially for the agar diffusion technique.

Point 3:

Concerning the results, the main points to be addressed are:

1.     Indicate a numerical scale on Y axis of Figures 1, 3 and 4

2.     In Figure 1, could be helpful to insert the standard XDR profile of Ag2O in the graph

3.     In Figure 2, plain PET fabric FESEM imaging is missing and must be provided.

4.     Regarding FESEM analysis, a comment and an explanation of the possible reason of the almost spherical form of Ag nanoparticles in sample F5 (pH 10,5) should be given

5.     In Figure 3, indications of the most relevant wavenumbers might be insert close to the peak or with an arrow indicating the exact wavenumber

6.   In Figure 4 and relative explanation in the text, increased adsorption is reported at wavelengths between 230 and 237 nm. There is total lack of the description and discussion of very broad but higher peak at around 410nm, which is normally due to the presence of silver nanoparticles. Furthermore, the spectrum of sample F1 highlights that the material is probably less decorated with Ag. Also this point deserve appropriate discussion. Finally, enlargement of the spectra in the most interesting part may be helpful (e.g. between 200 nm and 1000nm) excluding the last part above 1000 nm

7.     Regarding wettability, it would be helpful to acknowledge that the cut-off to indicate hydrphobicity or hydrophilicity can be also at 90° (e.g. Law KY, 2014; Ahmad I & Kan CW, 2016)

8.     Antibacterial test misses both positive (e.g. H2O2) and negative (plain PET fabric) control. Please provide them. Also information of how many replicates were performed is not reported and must be included. To strengthen antibacterial analysis a test on a Gram positive (Bacillus subtilis, Pseudomonas aeruginosa or Staphylococcus aureus) should be performed.

Point 4:

Line by line comments:

Lines 60-63: rephrase and refer to reference works in the field of nanoparticles (e.g. Ragelle et al., 2017; Torchilin, 2014);

Line 69: rapidity instead of fast

Line 81: mole

Line 85: consisting

Line 97: which strain of E. coli was used? Which broth –product number-?

Line 98: 1 mg/ml of which solution?

Line 103: in agar disc diffusion test, a lawn should be observed, not colonies.

Line 104: cite ImageJ with its appropriate reference

Line 117: sharp and narrow not broad and narrow

Line 121: insert citations of other groups

Line 134: literature not literatures

Lines 140-141: rephrase the sentence

Line 163: F5 at the end

Line 165: can you please clarify “information of materials information”?

Lines 192-193: literature not literatures; rephrase

Line 200: more instead of highly

Lines 201-205: It would be appropriate to tone down these sentences about self-cleaning since it is not demonstrated

Lines 214-222: rephrasing in view of the new literature suggested (Pelgrift & Friedman, 2013; Mijnendonckx et al., 2013; Lemire et al., 2013) is encouraged. Change of references 45 and 46 to more appropriate ones is also recommended. An appropriate reference for line 219 should be inserted. The explanation of what ATP is should not be inserted but more appropriate would be the explanation of the acronym.

Author Response

The introduction highlights the need for reusable cloth   that displays antibacterial activity especially for clinical settings, where   multidrug resistant strains are spreading. Then metals use as antimicrobial   agents is introduced. As for other parts, the authors should carefully revise   and thoroughly improve biological aspects. Concerning the introduction, an   improvement in the papers cited is suggested. More relevant research in the   field should be acknowledged (e. g. Levy & Marshall, 2004; O’Neill, J.   Tackling Drug-Resistant Infections Globally: Final Report and   Recommendations; Pelgrift & Friedman, 2013; Mijnendonckx et al., 2013;   Lemire et al., 2013).

Thanks for the suggestions. The following references has been   added to the article

(e. g. Levy & Marshall, 2004; O’Neill, J. Tackling Drug-Resistant   Infections Globally: Final Report and Recommendations; Pelgrift &   Friedman, 2013; Mijnendonckx et al., 2013; Lemire et al., 2013).

1-      The Materials and Methods section   should be rephrased in a clearer way in its whole, especially for the agar   diffusion technique.

Arthurs reply:

Thanks for the suggestions. We have amended the statements   as follows:

 “Disk diffusion   antibiotic sensitivity testing (Kirby–Bauer antibiotic testing) was employed   to evaluate the efficiency of PETs against Escherichia coli. Firstly, an   aqueous solution with 1 mg/mL concentration of the nanomaterial was prepared   all the solutions were sonicated to obtain a homogenous distribution of   nanomaterial in the solution. Subsequently, using a micro pipet 20 μL of the   prepared solution was added to the fabrics that were cut as disks, the   fabrics containing the nanoparticles were left at room temperature to dry   overnight. Then, 30 μL of E. coli solution, with a concentration of 16×107cells/mL   (OD600 of 0.2), was spread on a petri dish containing nutrient agar. The   fabric discs containing PETs nanomaterial were placed on each Petri dish. the   Petri dishes were incubated at 35 °C for 24 h,  digital images of the plates were captured,   and the inhibition zone was calculated in an image processing software (image   J.140) each test was repeated for 3 times and the average results was   reported.”

Point 3:

Concerning the results, the main points to be addressed are:

1.           Indicate a   numerical scale on Y axis of Figures 1, 3 and 4

Arthurs reply:

Thank you so   much for highlighting this issue numerical scale on Y axis of Figures 1, 3   and 4 has been fixed

In Figure 1, could be helpful to insert the standard XDR profile   of Ag2O in the graph   Arthurs reply:

Thanks for this constructive suggestion. The standard was added   to the Fig.1

2.           In Figure 2,   plain PET fabric FESEM imaging is missing and must be provided.

Arthurs reply:

Thank you so much for highlighting this issue the SEM Image was   added.

4.     Regarding FESEM analysis, a   comment and an explanation of the possible reason of the almost spherical   form of Ag nanoparticles in sample F5 (pH 10,5) should be given

Arthurs reply:

However, the rod-shaped particles were shortened to semi-spherical due   to lower Gibbs free energy and became homogenously dispersed, as shown in   Fig. 2j.

5.     In Figure 3, indications of the   most relevant wavenumbers might be insert close to the peak or with an arrow   indicating the exact wavenumber

Arthurs reply:

Thank you for your suggestion. Some of wavenumbers were   added due to control of crowding. 

6.   In Figure 4 and relative explanation in the   text, increased adsorption is reported at wavelengths between 230 and 237 nm.   There is total lack of the description and discussion of very broad but   higher peak at around 410nm, which is normally due to the presence[em1]  of silver nanoparticles.   Furthermore, the spectrum of sample F1 highlights that the material is   probably less decorated with Ag. Also this point deserve appropriate   discussion. Finally, enlargement of the spectra in the most interesting part   may be helpful (e.g. between 200 nm and 1000nm) excluding the last part above   1000 nm

Arthurs reply:

The maximum change was observed at wavelengths between 230   and 237 with an absorbance ~1.2, in comparing with the plain sample. As can   be observed, the maximum absorbance at the wavelength 410nm is ~1.1.So, I respectfully disagree with this suggestion

7.     Regarding wettability, it would   be helpful to acknowledge that the cut-off to indicate hydrphobicity or   hydrophilicity can be also at 90° (e.g. Law KY, 2014; Ahmad I & Kan CW,   2016)

Arthurs reply:

Thank you for your suggestion. 274 papers used  65° as an index for  hydrphobicity and  hydrophilicity based on google scholar.

https://scholar.google.com/scholar?hl=en&as_sdt=0%2C5&q=Hydrophobicity++%22contact+angle+65%22&btnG=

8. Antibacterial test misses both positive   (e.g. H2O2) and negative (plain PET fabric) control.   Please provide them. Also information of how many replicates were performed   is not reported and must be included. To strengthen antibacterial analysis a   test on a Gram positive (Bacillus subtilis, Pseudomonas   aeruginosa or Staphylococcus aureus) should be performed.

Arthurs reply:

·           Thanks   for this constructive suggestion. We understand this concern of the reviewer   and tried our very best to provide such information. As a matter of fact, we   have done control test for all the Petri dishes but the pictures that you   chose for the reporting didn't contain the control due to the fact that   authors wanted to provide a clean looking antimicrobial image that prevents   any confusion.

·           As   for the repetition, Authors would like to apologize for the confusion caused   due lack of explanation the disk diffusion tests were repeated for 3 times   and the average was reported.

 Amendment:

Materials and Methods page 3 line 107 to 108

·           Thanks   for this constructive suggestion. We sincerely wish to apologize for not be   able to provide more composition studies with gram-positive bacteria.   However, we would like to consider the suggestion for our near future work.   On the other hand, the fact that gram-negative bacteria’s such as E. coli   usually have higher resistance toward the antibiotics, and our material   showed reasonably high inhibition zone against the gram-negative E. coli we   think it would perform almost similar toward gram-positive bacteria. However,   we will definitely take your advice and perform the test in our near-future   work. 

Point 4:

Line by line comments:

Lines 60-63: rephrase and refer to   reference works in the field of nanoparticles (e.g. Ragelle et al., 2017;   Torchilin, 2014);

Line 69: rapidity instead of fast

Line 81: mole

Line 85: consisting

Line 97: which strain of E. coli was used?   Which broth –product number-?

Line 98: 1 mg/ml of which solution?

Line 103: in agar disc diffusion test, a   lawn should be observed, not colonies.

Line 104: cite ImageJ with its appropriate   reference

Line 117: sharp and narrow not broad and   narrow

Line 121: insert citations of other groups

Line 134: literature not literatures

Lines 140-141: rephrase the sentence

Line 163: F5 at the end

Line 165: can you please clarify   “information of materials information”?

Lines 192-193: literature not literatures;   rephrase

Line 200: more instead of highly

Lines 201-205: It would be appropriate to   tone down these sentences about self-cleaning since it is not demonstrated

Lines 214-222: rephrasing in view of the   new literature suggested (Pelgrift & Friedman, 2013; Mijnendonckx et al.,   2013; Lemire et al., 2013) is encouraged. Change of references 45 and 46 to   more appropriate ones is also recommended[em2] . An appropriate   reference for line 219 should be inserted. The explanation of what ATP is   should not be inserted but more appropriate would be the explanation of the   acronym

Arthurs   reply:

Thank   you for your suggestion. The amendment was done.

Thank   you so much for this constructive comments the manuscript has been sent for   proofreading and the changes requested has been made.

Necessary   changes to the references have been made.

1-      the paper would benefit from better   organization. I felt that the order of presenting information was somewhat   confusing.?

Arthurs reply:

Thanks for highlighting this issue. Authors   have tried to improve the quality of the paper as much as possible most of   the explanations have been improved.

2-      the paper needs to be proofread.   There are few mistakes, for example line 165 "popular method to obtain   the information of materials information"

Arthurs reply:

Thanks   for this constructive suggestion. The paper was sent for proofreading.

Reviewer 3 Report

The work is timely and the methods ae sound. I recommend the following minor revision: 1- the paper would benefit from better organization. I felt that the order of presenting information was somewhat confusing. 2- the paper needs to be proofread. There are few mistakes, for example line 165 "popular method to obtain the information of materials information" 3- The authors synthesized the samples at different pH. For example F4 and F5 have relatively high pH compared to the other samples. When the authors did the toxicity experiments to E.coli, could the results also be affected by difference in pH and not only the characters of the nanoparticles? Does a basic pH affect optimal E. Coli growth?

Author Response

1-      The authors synthesized the samples at different pH. For example F4 and F5 have relatively high pH compared to the other samples. When the authors did the toxicity experiments to E.coli, could the results also be affected by difference in pH and not only the characters of the nanoparticles? Does a basic pH affect optimal E. Coli growth

Arthurs reply:

Thank you for rising up this issue. Authors would like to apologize for the confusion caused due lack of explanation, as a matter of fact after the fabric was coated in different pH they were washed and rinsed with ultrapure water hence the pH did not affect the bacterial growth in the toxicity test part.

Round  2

Reviewer 2 Report

The authors presented a revised manuscript in which a minority of my requests were properly addressed (e.g. Y-axis scales in Fig. 1, 3, 4 and SEM images of plain material).

Authors also write in their answers that new suggested literature was added. I unfortunately cannot find any trace of it in the new version of the manuscript provided. Further, the majority of corrections asked by me and other reviewers were not addressed at all ( one for all, " information of materials information", line 232 of the new manuscript). Other examples are the specific strain of E. coli used and the proper citation of ImageJ program.

Furthermore I understand authors' concers in keeping Figure 6 as clean as possible for clarity reasons. However, images of controls must be provided, even also as supplementary file. There is no trace of it in materials and method section, nor in the table with halo measures.

Author Response

Dear Editor, 

Thank you for your useful comments on the structure of our manuscript.

We have modified the manuscript accordingly, and the corrections are listed in the following . The imposed changes are in blue yellow color in original paper. I do appreciate the wise comment and try to satisfy each comment accordingly

Point 1: Authors also write in their answers that new suggested literature was added. I unfortunately cannot find any trace of it in the new version of the manuscript provided. Further, the majority of corrections asked by me and other reviewers were not addressed at all ( one for all, " information of materials information", line 232 of the new manuscript). Other examples are the specific strain of E. coli used and the proper citation of ImageJ program.

Response 1:

Thanks for the suggestions. We have amended the statements as follows:

Firstly, fabrics embedded with nanoparticles at different p.H. were a cut as disks with a diameter of 0.5 mm. Then, 30 μL of E. coli solution (nutrient broth Himedia M001 U.S.A.), with a concentration of 16 × 107 cells/mL (OD600 of 0.2), was spread on a petri dish containing nutrient agar (Himedia M002 U.S.A). The fabric discs containing Ag2O nanomaterial were placed on each Petri dish a piece of uncoated fabric was used as a control in each set of test. The Petri dishes were incubated at 35 °C for 24 h, digital images of the plates were captured, and the inhibition zone was calculated using an image processing software (image J.140) [20,30] each test was repeated for 3 times and the average results was reported.

Point 2: Furthermore I understand authors' concers in keeping Figure 6 as clean as possible for clarity reasons. However, images of controls must be provided, even also as supplementary file. There is no trace of it in materials and method section, nor in the table with halo measures.

Response 2:

Thanks for this constructive suggestion. The supplementary file  was added at the end of manuscript.

Round  3

Reviewer 2 Report

Authors have now answered to my previous points.